# *Lactiplantibacillus plantarum* A72, a Strain with Antioxidant Properties, Obtained through ARTP Mutagenesis, Affects *Caenorhabditis elegans* Anti-Aging

**DOI:** 10.3390/foods13060924

**Published:** 2024-03-19

**Authors:** Sibo Zou, Qi Wu, Zhigao Li, Sufang Zhang, Liang Dong, Yingxi Chen, Yiwei Dai, Chaofan Ji, Huipeng Liang, Xinping Lin

**Affiliations:** SKL of Marine Food Processing & Safety Control, National Engineering Research Center of Seafood, Collaborative Innovation Center of Provincial and Ministerial Co-Construction for Deep Processing, Collaborative Innovation Center of Seafood Deep Processing, School of Food Science and Technology, Dalian Polytechnic University, Dalian 116034, China; 211710832000939@xy.dlpu.edu.cn (S.Z.); 221720860001022@xy.dlpu.edu.cn (Q.W.); 211710832000932@xy.dlpu.edu.cn (Z.L.); zhangsf@dlpu.edu.cn (S.Z.); dongliang@dlpu.edu.cn (L.D.); chenyx@dlpu.edu.cn (Y.C.); daiyiwei@dlpu.edu.cn (Y.D.); jichaofan@outlook.com (C.J.); lhpdxx@126.com (H.L.)

**Keywords:** *Lactiplantibacillus plantarum*, *Caenorhabditis elegans*, antioxidants, anti-aging, transcriptome sequencing

## Abstract

This research endeavored to elucidate the antioxidant attributes of lactic acid bacteria, specifically their impact on anti-aging and lifespan augmentation in *Caenorhabditis elegans*. The study focused on *Lactiplantibacillus plantarum* A72, identified through ARTP mutagenesis for its potent antioxidant properties. In vitro analysis affirmed its free radical neutralizing capacity. In *C. elegans*, the strain not only extended the lifespan by 25.13% and amplified motility 2.52-fold, but also maintained reproductive capabilities. Remarkably, *Lpb. plantarum* A72 diminished reactive oxygen species (ROS) and malondialdehyde (MDA) levels in *C. elegans* by 34.86% and 69.52%, respectively, while concurrently enhancing its antioxidant enzyme activities. The strain also bolstered *C. elegans* survival rates by 46.33% and 57.78% under high temperature and H_2_O_2_ conditions, respectively. Transcriptomic scrutiny revealed that *Lpb. plantarum* A72 could retard *C. elegans* aging and extend lifespan by upregulating the *sod-5* and *hsp-16.1* genes and downregulating the *fat-6* and *lips-17* genes. These findings propose *Lpb. plantarum* A72 as a potential antioxidant and anti-aging lactic acid bacteria.

## 1. Introduction

Aging is a complex biological process, mediated by both genetic and environmental factors [1]. In most cases, aging-related diseases and programmed cell death are primarily caused by cellular oxidative stress [2]. In order to ameliorate the aging-related diseases that accompany the process of increasing human longevity, the focus is increasingly on promoting a healthy lifespan based on slowing down aging. Improving the body’s oxidative stress capacity through diet is an effective way to fight aging.

There is mounting evidence that naturally occurring active microorganisms like lactic acid bacteria (LAB) possess potent antioxidant activity. These microorganisms have probiotic potential, including intestinal flora balance, body immunity regulation, anti-cellular senescence and antioxidant activity [3,4]. Among LAB, one of the more versatile and promising species is *Lactiplantibacillus plantarum*. Using *Lpb. plantarum* J26 to ferment blueberry juice, Zhang et al., 2021 discovered that fermentation had the effect of reducing oxidative damage in a Caco-2 cell model and greatly increased the scavenging ability of DPPH, superoxide anion radicals, and hydroxyl radicals [5]. Pan found that *Lpb. plantarum* KFY02 was able to convert geniposide into genipin, which reduced CCl_4_-induced liver injury in mice through the ability to enhance the antioxidant capacity of the organism. Although the antioxidant effects of *Lpb. plantarum* microbiota have been demonstrated in several studies, reports on its aging-delaying effects and related mechanisms are rare [6].

*Caenorhabditis elegans* is a model organism widely used in the field of biomedicine, with the advantages of a short life cycle, a transparent body and ease of observation [7]. Its unique characteristics have made it the perfect model for a variety of studies: its small size for simple culture, its transparent body for simple labeling and observation, hermaphroditism for high fertility and experimental embryos or offspring samples, a short life cycle for quick lab research, and a distinct genetic background for comprehending referential mechanisms. In recent years, *C. elegans* has been widely used to study diseases related to antioxidation, anti-aging, etc. Li et al., 2022 investigated the antioxidant capacity of *Laminaria japonica* polysaccharide with *C. elegans*, and found that polysaccharides could enhance the expression of antioxidant enzymes and reduce the levels of malondialdehyde (MDA) and reactive oxygen species (ROS) in *C. elegans* [8]. Sun et al., 2023 investigated the anti-aging effects of polysaccharides extracted from ginsenoside residues (GRP) with *C. elegans*, and found that polysaccharides could enhance the antioxidant activity in *C. elegans*, thereby avoiding oxidative damage and extending their lifespan [9]. However, the specific antioxidant, anti-aging, and lifespan extension mechanisms of microorganisms (e.g., LAB) on *C. elegans* are unclear.

In this paper, *Lpb. plantarum* A72 with antioxidant capacity was screened by Atmospheric and Room Temperature Plasma (ARTP) mutagenesis. After that, we used *C. elegans* as a model to evaluate the effects of the molecular mechanism of *Lpb. plantarum* A72 extract on anti-aging. Specifically, we studied the physiological conditions, resistance to environmental factors, antioxidant enzyme activities and transcriptome sequencing of *C. elegans*. Eventually, the mechanism for the lifespan extension of *C. elegans* with the addition of *Lpb. plantarum* A72 was discussed. Our present investigation may accelerate the utilization of *Lpb. plantarum* A72 as a potential dietary supplement to retard age-related diseases and enhance health span.

## 2. Materials and Methods

### 2.1. Materials and Reagents

*Lpb. plantarum* SC3 was selected from Chinese fermented vegetables (Dalian, China) and was stored in our lab [10]. *Lpb. plantarum* SC3 was used as the original strain and the ARTP (ARTP Mutagenesis Breeding Machine, ARTP-IIS, Tmaxtree, Wuxi, China) technique was used to generate mutant strains with high antioxidant capacity. After hydrogen peroxide tolerance evaluations, *Lpb. plantarum* A72 with the highest antioxidant capacity was selected from 149 strains. *Caenorhabditis elegans* N2 (wild-type) and *Escherichia coli* strain OP50 were purchased from SunyBiotech (Fuzhou, China).

Total superoxide dismutase (T-SOD), catalase (CAT) and malondialdehyde (MDA) kits were purchased from the Nanjing Jiancheng Institute of Biological Engineering (Nanjing, China).

### 2.2. Preparation of Cell-Free Supernatant (CFS), Intact Cells (IC), and Cell-Free Extracts (CFE) from Lpb. plantarum

*Lpb. plantarum* was cultivated in MRS broth at 37 °C for 24 h and adjusted to 10^9^ CFU/mL. Following a 10 min centrifugation at 4 °C and 6000 rpm, the supernatant was measured and recorded as CFS. The precipitate was resuspended after being washed three times with 0.85% (*w*/*v*) saline; one portion was labeled as IC, and the other was broken up by ultrasonication, centrifuged, and the supernatant was removed and labeled as CFE.

### 2.3. Free Radical Scavenging Capacity Assay of Lpb. plantarum

The 2,2-diphenyl-1-picrylhydrazyl (DPPH)-scavenging activities, superoxide anion radical-scavenging activities, hydroxyl radical-scavenging activities and the 2-phenyl-4,4,5,5-tetramethyl-2,5-dihydro-1H-imidazol-1-oxyl 3-N-oxide (PTIO)-radical-scavenging activities of *Lpb. plantarum* SC3 and *Lpb. plantarum* A72 were analyzed as described by Wang et al., 2022 [11], Li et al., 2021 [12], Li et al., 2021 [13] and Li et al., 2017 [14], respectively. As a control, 0.1 mg/mL Vitamin C (VC) solution was prepared. All experiments were performed in triplicate.

### 2.4. C. elegans Culture Conditions

The *C. elegans* were lysed with a ready-to-use cell lysis solution [15], then centrifuged at 6000 rpm for 1 min to obtain the eggs, and finally washed with M9 buffer to exhaust the cell lysate. The eggs were transferred to Nematode Growth Medium (NGM) and incubated at 25 °C for 48 h and then developed into L4 stage larvae.

*E. coli* OP50 was cultured and suspended in saline to OD_600_ = 0.6. *Lpb. plantarum* A72 resuspended in the above *E. coli* OP50 solution to 10^5^, 10^7^ and 10^9^ CFU/mL were used as the experimental group. VC was diluted in OP50 solution to 0.1 mg/mL and used as the positive control. The OP50 resuspension group was used as a negative control group. For each group, 200 μL of the solution was spread on NGM medium as *C. elegans* food for three parallels, and cultivated at 25 °C [15].

### 2.5. Effects of Lpb. plantarum A72 Addition on the Physiological Conditions of C. elegans

#### 2.5.1. Effect of *Lpb. plantarum* A72 Addition on the Lifespan of *C. elegans*

Thirty L4 stage *C. elegans* were transferred to NGM dishes and placed in an incubator at 25 °C to measure their lifespan according to Zhao et al., 2023 [15]. The survival of *C. elegans* was counted every 24 h from the date of transfer to a new NGM dish.

In the absence of any response from *C. elegans*, the head of *C. elegans* was touched and the number of *C. elegans* deaths was counted until all *C. elegans* were dead.

#### 2.5.2. Effect of *Lpb. plantarum* A72 Addition on the Reproductive Capacity of *C. elegans*

The reproductive capacity was carried out with L4 stage *C. elegans* on NGM (five per plate) [16]. They were changed to a new medium to ensure adequate nutrients for every 24 h, and the process continued for 7 days. The number of *C. elegans* eggs on the media was counted each day. The number of progeny was recorded as the total number of eggs per *C. elegans*.

#### 2.5.3. Effect of *Lpb. plantarum* A72 Addition on the Head Swing of *C. elegans*

The head swing of *C. elegans* was used to determine its mobility, and its head swing was quantified in accordance with Wang et al., 2023 [17]. After 48 h of administration, the number of head swings was counted for 30 s each time. The experiment was repeated three times with at least 30 *C. elegans* per group.

### 2.6. Resistance of C. elegans to Environmental Factors with Lpb. plantarum Supplement

The effect of *Lpb. plantarum* A72 on *C. elegans* resistance to environmentally induced oxidative damage was measured according to Wang’s study [18].

#### 2.6.1. Effect of *Lpb. plantarum* on the Survival Rate of *C. elegans* under Thermal Shock

The L4 *C. elegans* in each group were cultured for 48 h, and then transferred to an incubator at 35 °C. After heat treatment for 4 h, the survival rate was calculated. The heads of *C. elegans* were touched and if there was no response, *C. elegans* were considered dead.

#### 2.6.2. Effect of *Lpb. plantarum* on the Survival Rate of *C. elegans* Stimulated by H_2_O_2_

The L4 *C. elegans* in each group were cultured for 48 h, and then transferred to NGM petri dishes containing 10 mM H_2_O_2_ and cultured for 4 h at 25 °C. After that, the survival rate was calculated. *C. elegans*’ death was judged as previously described.

### 2.7. Determination of the Antioxidant Enzyme Activities, MDA Levels and ROS Accumulation of C. elegans with Lpb. plantarum Supplement

The *C. elegans* were washed three times with M9 buffer then crushed by an ultrasonic wave to obtain a suspension. The total superoxide dismutase (T-SOD), catalase (CAT) and malondialdehyde (MDA) levels of *C. elegans* were measured according to the kits’ instructions. The ROS accumulation of *C. elegans* was measured as described by Li et al. [19] with infinite F200 PRO (TECAN, Zürich, Switzerland).

### 2.8. Transcriptome Sequencing of C. elegans

Total RNA was extracted from at least 2000 *C. elegans* samples with an MJZol total RNA extraction kit (Majorbio, Shanghai, China). The concentration and purity of the extracted RNA were examined using Nanodrop2000 (Thermo Fisher Sci, Pleasanton, CA, USA), and after the quality was qualified, the library was constructed using Illumina^®^ Stranded mRNA Prep, Ligation from Illumina (San Diego, CA, USA) and sequenced on the Illumina NovaSeq 6000 (Illumina, CA, USA) platform.

Adapters were removed from the obtained data, and clean data were obtained after removing the low-quality reads, and comparing them with the reference genome (Caenorhabditis elegans WBcel235: https://metazoa.ensembl.org/Caenorhabditis_elegans/Info/Index (accessed on 1 July 2023). The expression levels of the genes were quantified using the software RSEM (http://deweylab.github.io/RSEM/ (accessed on 1 July 2023) and processed by DEGseq (https://www.rdocumentation.org/packages/DEGseq (accessed on 1 July 2023) to screen for significantly differentially expressed genes, setting the default screening criteria as FDR < 0.05 & |log2FC| ≥ 1. Differential genes were analyzed by means of the Gene ontology (GO) (http://geneontology.org/ (accessed on 1 July 2023) and Kyoto encyclopedia of genes and genomes (KEGG) (https://www.genome.jp/kegg/ (accessed on 1 July 2023) enrichment.

### 2.9. Data Analysis

Graphs were created with Origin 2021 (OriginLab Corporation, Northampton, MA, USA). The analysis of significant differences among samples was carried out using SPSS Statistics 26 (SPSS Inc., Chicago, IL, USA).

## 3. Results and Discussion

### 3.1. Antioxidant Activity In Vitro of Lpb. plantarum A72

*Lpb. plantarum* SC3 was screened from naturally fermented Suancai with in vitro antioxidant capacity equivalent to 270.58 μmol/L *L*-cysteine [10]. To further enhance the microbiota antioxidant capacity, ARTP mutagenesis was used. ARTP mutagenesis breeding technology is characterized by its simple operation, safety, environmental friendliness, high mutation rate and large mutation reservoir. Therefore, it has been widely used in the field of food and microbiology [19]. A total of 149 mutants were isolated after ARTP treatment (Appendix A) and cultured in MRS broth containing 1.5 mM H_2_O_2_ (Appendix A). The study’s findings led to the identification of *Lpb. plantarum* A72 as the best mutagenized strain. In vitro antioxidant activity was then compared to that of *Lpb. plantarum* SC3, the original strain. The growth curve of *Lpb. plantarum* A72 was examined, along with its tolerance to acid, bile salt, and gastric and intestinal fluids. The findings demonstrated that *Lpb. plantarum* A72 exhibited survival rates of 75.58%, 69.12%, 71.93% and 83.63%, respectively, in MRS broth at pH 3.0, MRS broth with bile salt content of 0.3% (*w*/*v*), artificial gastric fluid and artificial intestinal fluid (Appendix A). 0–9 mM H_2_O_2_ was shown to be conducive to the growth of *Lpb. plantarum* A72 (Appendix A). These findings suggest that *Lpb. plantarum* A72 is a strain that can grow in H_2_O_2_ and is tolerant to acid, bile salts, intestinal fluid and gastric fluid.

To study the antioxidant activity of A72, the original strain SC3 and different parts of the strain, we prepared cell-free supernatant (CFS), intact cells (IC) and cell-free extract (CFE), respectively. The IC and CFS of both strains showed greater DPPH radical scavenging ability than CFE (*p* < 0.05) (Figure 1A), with inhibition rates of 60.14% and 59.65%, respectively. The IC and CFS of *Lpb. plantarum* A72 also had strong scavenging activity of superoxide anion radicals (Figure 1B). The superoxide anion scavenging rate of the IC was 41.93%, similar to that of VC. Furthermore, the IC of both strains was significantly greater than that of CFS (*p* < 0.05). Luan et al., 2021 also found that the IC of *Lpb. plantarum* CD101 had a significantly greater scavenging capacity of superoxide anion than CFS (*p* < 0.05), which may be related to intracellular antioxidant substances (antioxidant enzymes, VC, etc.) [20]. The IC or CFS of both strains had higher hydroxyl radical scavenging activity than VC (Figure 1C). CFS had higher hydroxyl radical scavenging activity compared to IC and CFE, and the CFS of *Lpb. plantarum* A72 had the highest hydroxyl radical scavenging activity of 87.01%, which was 1.10 times higher than that of *Lpb. plantarum* SC3. Noureen et al., 2018 also found the highest scavenging capacity of 62.75% for hydroxyl radicals by the CFS of *Levilactobacillus brevis* MG882399 in their study [21]. Studies have shown that *Lactobacillus* sp. can produce a variety of extracellular compounds such as organic acids, alcohols, phenols, extracellular polysaccharides, bacteriocins and bioactive peptides through fermentation. These compounds may have antioxidant properties [22]. Similarly, in terms of PTIO radical scavenging activity (Figure 1D), both strains showed weaker viability of CFE and IC than CFS. The PTIO radical scavenging activity of IC or CFS of A72 was higher than that of SC3, and comparable to or even higher than that of VC. All fractions of *Lpb. plantarum* A72 showed greater scavenging capacity for all free radicals than *Lpb. plantarum* SC3. A study of antioxidant *Lpb. plantarum* ZJ316 by Wu et al., 2023 also revealed that the antioxidant may be strain-specific [23]. The high antioxidant property of *Lactobacillus* may be related to the antioxidant function-related genes it possesses, such as the NADH oxidase gene (*nox*), NADH peroxidase gene (*npx*) and GSH reductase gene (*gshR*). In conclusion, the *Lpb. plantarum* A72 strain obtained from the screening had strong antioxidant properties, and its IC and CFS components were significantly stronger than VC, and also stronger than its initial strain SC3, which was strain-specific.

### 3.2. Effects of Lpb. plantarum A72 Intact Cells Supplementation on C. elegans Physiological Conditions

As shown in Figure 2A and Table 1, the average lifespan of *C. elegans* in the control group was 25.33 ± 0.47 days. After intervention with VC or *Lpb. plantarum* A72, the average lifespan and maximum lifespan of *C. elegans* were dose-dependently increased (*p* < 0.05). The highest-dose (10^9^ CFU/mL) *Lpb. plantarum* A72 was the most effective in prolonging the lifespan of *C. elegans*, with a 25.13% an increase compared to the control. Li et al., 2022 similarly found that in a *C. elegans* ageing model, the application of *Lpb. plantarum* As21 increased the *C. elegans* lifespan [24]. In another study by Sharma et al. (2018), they intervened with 15 probiotics on *C. elegans* [25]. The results showed that *Lcb. paracasei* CD4, *Lim. gastricus* BTM7 and *Lpb. plantarum* K90 enhanced the lifespan by 5 days. They believed that the probiotics and postbiotics increased the lifespan of *C. elegans* via adherence and further colonization of *C. elegans* in the gut.

Some studies have shown that a longer lifespan was accompanied by a decline in fertility [26]; therefore, we explored the egg production of *C. elegans*. The results, as shown in Figure 2B, showed no significant effect on the total number of offspring in each group compared with the control group, indicating that the addition of *Lpb. plantarum* A72 did not affect the fecundity of *C. elegans* and did not induce the extension of its lifespan by sacrificing their reproduction. A study by Cai et al., 2022 also showed that there was no absolute relationship between the extension of *C. elegans* lifespan and the decrease in fecundity [27]. In their study, it was found that at a concentration of rice bran peptide KF-8 of 0.1 mM, *C. elegans* lifespan was extended by 17%, but *C. elegans* fecundity was not affected. During the ageing process, *C. elegans* loses muscle cell activity, leading to a reduction in its motility [28]; therefore, head swing was used as an evaluation index of *C. elegans*. As shown in Figure 2C, all the supplemented groups increased the head swing compared with the control group. *C. elegans*’ head swing was even significantly stronger (*p* < 0.05) in the groups supplemented with 10^7^ CFU/mL and 10^9^ CFU/mL *Lpb. plantarum* A72 than in the VC group. In previous studies, it was also found that the head swing of *C. elegans* was about 1.18 times higher than that of the control group, in which *C. elegans* were fed with 1.25 mg/mL of shiitake mushroom polysaccharides [29]. In addition, the head swing of *C. elegans* was also about 1.22 times higher than the control when fed with 0.1 mM ice bran peptide KF-8.26. This suggests that the addition of *Lpb. plantarum* (especially A72) helps to resist senescence and maintain head swing in *C. elegans*.

### 3.3. Effect of Lpb. plantarum A72 Intact Cells on C. elegans Resistance to Environmentally Induced Oxidative Damage

Environmental factors can induce oxidative damage in the organism [30].

The high temperature induction and hydrogen peroxide induction were used to test *C. elegans* resistance. The survival rate of *C. elegans* in the control group was 45.56% in a high-temperature heat stress test (Figure 3A), while the survival rate increased by 29.26% and 46.33% in *Lpb. plantarum* A72 addition groups (10^7^ CFU/mL and 10^9^ CFU/mL), respectively, which were 1.10 and 1.25 times higher than in the groups supplemented with VC. Similarly, it was found in a previous study that both 0.8 mg/mL and 1.6 mg/mL ethanol extracts of Panax notoginseng (FPE) significantly increased the longevity of *C. elegans*, by 25.30% and 40.05%, respectively, at 37 °C [31].

In the hydrogen peroxide oxidative stress environment (Figure 3B), the survival rate of *C. elegans* in the control group was only 35.56%. A significant increase in *C. elegans* survival was observed with the addition of VC or *Lpb. plantarum* A72 (*p* < 0.05). *C. elegans* survival increased to 40.00%, 45.56% and 57.78% in the low, medium and high *Lpb. plantarum* A72 dose groups, respectively. The effect of the low- and medium-dose groups was comparable to that of VC, while the high-dose group (10^9^ CFU/mL) was significantly better than VC. This indicated that there was a significant dose–effect relationship between the addition of *Lpb. plantarum* A72 and the survival of *C. elegans*. In Jin’s study, *C. elegans* was fed with 0.8 and 1.6 mg/mL of ethanol extract (Panax notoginseng FPE), and it was found that the fed *C. elegans* could survive in a hydrogen peroxide environment and there was a significant dose–effect relationship [31]. Among them, the lifespan of *C. elegans* was enhanced by 25.77% and 35.88% in the low- and high-concentration groups of FPE compared with the control group, respectively. In contrast, our *Lpb. plantarum* A72 was more resistant to oxidative stress in *C. elegans* at each concentration than the high-concentration FPE.

Overall, our study found that the *Lpb. plantarum* A72 strain exhibited similar properties to the FPE in *C. elegans*, especially in terms of heat tolerance and resistance to oxidative stress. Therefore, the tolerance of *C. elegans* to high temperatures and oxidative stress could be improved by supplying feeds enriched with an antioxidant strain (*Lpb. plantarum* A72).

### 3.4. Determination of the ROS Accumulation, Antioxidant Enzyme Activities and MDA Levels in C. elegans

As one of the most important oxidants, the level of reactive oxygen species (ROS) can indirectly reflect the degree of oxidation of the body’s cells, and its accumulation can cause oxidative damage to the body [32]. As shown in Figure 4A, low, medium and high concentrations of *Lpb. plantarum* A72 were able to reduce the ROS levels and thus protect *C. elegans* from oxidative stress compared to the control group (*p* < 0.05). In addition, the ability of *Lpb. plantarum* A72 to inhibit ROS was related to its concentration. The addition of the 10^5^ CFU/mL A72 group only reduced ROS by 13.03%, while the addition of 10^7^ CFU/mL and 10^9^ CFU/mL A72 groups had a comparable ability to reduce ROS as VC, and in particular, the effect of the 10^9^ CFU/mL A72 group was even better than that of VC (*p* < 0.05). The results indicated that *Lpb. plantarum* A72 had good antioxidant properties and could replace VC at a concentration of 10^9^ CFU/mL. Lu et al., 2023 indicated that *Latilactobacillus curvatus* FFZZH5L might have achieved its antioxidant effect by reducing ROS in *C. elegans*, for example, by regulating the genes (*gst-4*) related to ROS in its body [33]. Differences in the antioxidant properties of the two strains of *Lpb. plantarum* were also found in our study, namely that the ROS content of *C. elegans* treated with 10^9^ CFU/mL *Lpb. plantarum* As21 and As39 were reduced by 45.78% and 29.95%, respectively. In an investigation by Jin et al., 2020 variations in the antioxidant resistance of several strains were also discovered [34]. When they fed *C. elegans* with 10 *Lactobacillus*, they observed that the most degrading strain, *Limosilactobacillus reuteri* 9-5, broke down 30% of the reactive oxygen species (ROS).

SOD scavenges superoxide anion radicals produced by the body [35]. CAT could degrade or reduce hydrogen peroxide to water and molecular oxygen [35]. The activities of SOD and CAT can reflect the antioxidant capacity of *C. elegans*. In Figure 4B,C, the activities of SOD and CAT were increased by the addition of *Lpb. plantarum* A72, and were found to be positively related with the concentration of *Lpb. plantarum* A72. Among them, the 10^9^ CFU/mL A72 group had the best effect in increasing the activities of SOD and CAT, which were 1.49 and 1.59 times higher than those of the control group and 1.36 and 1.10 times higher than those of the VC group, respectively. These results are consistent with the above results that ROS accumulation in *C. elegans* (Figure 4A) thus induced higher expression of SOD and CAT enzymes resulting from the stress oxidative responses. Rhodiola rosea extract is a substance with high oxidizing activity. In a previous study, Jiang et al., 2023 fed 480 μg/mL of Rhodiola rosea extract to *C. elegans*, which resulted in a 1.78-fold and 1.51-fold increase in SOD and CAT activities, respectively [36]. In addition, Li et al., 2022 found that the SOD and CAT activities in *C. elegans* increased 1.31-fold and 1.74-fold, respectively, after 10^9^ CFU/mL *Lpb. plantarum* As21 was fed to *C. elegans* [24]. In our result, SOD activity in *C. elegans* was higher after ingestion of the same concentration of *Lpb. plantarum* A72 than after ingestion of *Lpb. plantarum* As21. These results suggest that the antioxidant capacity of *Lpb. plantarum* A72 obtained from the screening is comparable to that of Rhodiola rosea extracts and even superior to other strains reported in the current literature.

Tissue damage by oxygen radicals leads to the production of MDA, and the level of MDA in the body can indirectly reflect the degree of oxidative damage in the body [36]. After the treatment of *Lpb. plantarum* A72 (Figure 4D), the MDA content in *C. elegans* was significantly lower than that in the control group and also lower than that in the VC group, which indicated that *Lpb. plantarum* A72 had a significant effect on the reduction of MDA content, even more so than that of VC. The reduction of MDA by *Lpb. plantarum* A72 was enhanced with the increase of the concentration. When 10^9^ CFU/mL of *Lpb. plantarum* A72 was added, the reduction of MDA level in *C. elegans* could reach 69.52% (*p* < 0.05). In previous research, Li et al., 2022 reduced the MDA content of *C. elegans* by 61.14% with 10^9^ CFU/mL *Lpb. plantarum* As39 [24], and Jiang et al., 2021 reduced the MDA content of *C. elegans* by 63.4% after feeding with 480 μg/mL Rhodiola rosea extract [28]. The results of A72 in our study were better than those previous reports, indicating that the strain was effective in reducing lipid peroxidation damage in *C. elegans*.

In summary, the *Lpb. plantarum* A72 strain can realize its antioxidant function by reducing ROS and MDA levels in *C. elegans* and enhancing SOD as well as CAT activity in *C. elegans*. In order to investigate the specific mechanism, we conducted transcriptomic studies on *C. elegans* that ingested A72.

### 3.5. Differential Gene Expression of C. elegans Induced by Lpb. plantarum A72 Addition

Compared with the control group, the number of upregulated genes in the A72 group was 58, and the number of downregulated genes was 72, which showed obvious differences in transcript levels (Figure 5A). The differential genes were annotated and analyzed by means of the GO database (Figure 5B), and the differential genes were enriched in terms of cellular components (red), molecular functions (blue), and biological processes (green) in the control and A72 groups. KEGG annotation analysis showed that the differentially expressed genes between the two groups were annotated to the following six branches of the KEGG metabolic pathway: metabolism, genetic information processing, environmental information processing, cellular processes, organismal systems and others. Notably, more differentially expressed genes were annotated to the organismal systems branch between the two groups (Figure 5C). This suggests that the uptake of the *Lpb. plantarum* A72 strain may delay aging by affecting the expression of aging-related genes in *C. elegans*.

### 3.6. Mechanistic Investigation of C. elegans Lifespan Extension by Lpb. plantarum A72 Supplement

KEGG pathway analysis found that *sod-5* was significantly upregulated after the consumption of *Lpb. plantarum* A72 compared to the control *C. elegans* without the *Lpb. plantarum* A72 supplement (Figure 6A). The upregulation of the *sod-5* gene promoted the synthesis of superoxide dismutase (SOD). A similar phenomenon was observed in a study by Lin et al., 2019, who found that consumption of the antioxidant Rosmarinic acid (RA) significantly extended the lifespan of *C. elegans*, while significant upregulation of the *sod-5* gene was also observed in *C. elegans* [37]. This is due to the fact that SOD is an enzyme that counteracts the biological oxidant cluster enzyme system and plays an important role in cellular oxidative stress, lipid metabolism, inflammation and oxidation. Expression of the SOD enzyme contributed to the scavenging of bio-oxidants in the body, thereby counteracting the harmful effects of free radicals and slowing down the aging of organisms [38]. In addition, we also observed a significant upregulation of the *hsp-16.1* gene, responsible for encoding the small heat shock proteins (sHSPs), in *C. elegans* after consumption of *Lpb. plantarum* A72 (Figure 6B). Qin fed *C. elegans* with Ligusticum chuanxiong rhizome extracts (CXR) and observed that the *C. elegans* lifespan was prolonged and that *hsp-16.1* genes were also significantly upregulated in nematodes. Organismal aging and aging-related degenerative diseases are associated with the misfolding and aggregation of many proteins in various tissues [39]. sHSPs was reported to slow down the aging of organisms and ultimately prolonged lifespan by binding to misfolded proteins and preventing their uncontrolled aggregation [40]. The anti-aging effects of both CXR and the consumption of *Lpb. plantarum* A72 may be related to the overexpression of sHSPs [41]. Interestingly, earlier research found a positive association between the heat shock transcription factor HSF-1 and the expression of the *hsp-16.1* gene [42], as well as a positive correlation between the transcription factor DAF-16 and the expression of the *sod-5* gene [43]. However, the gene of transcription factors (*daf-16* and *hsf-1*) did not show significant differential expression in our transcriptome data (Figure 6C,D). It suggests that the upregulation of two genes, *sod-5* and *hsp-16.1*, might not depend on the traditional DAF-16 and HSF-1-controlled pathways to realize the role of delaying senescence in *C. elegans*, and the specific mechanism remains to be further explored.

Following the administration of *Lpb. plantarum* A72, a notable observation surfaced: the significant downregulation of the *fat-6* gene (Figure 6E) within *C. elegans*. This gene serves as an inhibitor of fatty acid synthesis, a pivotal factor in cellular oxidative stress. The surplus accumulation of fatty acids precipitates oxidative stress, thereby impairing cellular functionality and expediting the biological aging process. Mitigating the buildup of fatty acids stands as a potential avenue for extending organismal lifespan, similar to the idea that reducing obesity will improve the general health of the organism. The parallel effects of downregulation were discerned in a previous study involving astaxanthin-fed *C. elegans*, where the intake of astaxanthin concomitantly reduced body fat content and prolonged the organism’s lifespan [44]. In light of this, our current investigation posits that the intake of *Lpb. plantarum* A72 might effectively extend the lifespan of *C. elegans* by orchestrating the downregulation of the *fat-6* gene, thereby impeding fatty acid synthesis.

Furthermore, our analysis showed that the *lips-17* gene, which is closely linked to the synthesis of fatty acids [45], was downregulated (Figure 6F). Our hypothesis suggests a possible connection, like to that of *fat-6*, between the downregulation of the *lips-17* gene and the elongation of the lifespan of *C. elegans*. Remarkably, this observed relationship between *lips-17* downregulation and lifespan extension represents a novel revelation from our investigation.

In summary (Figure 6G), transcriptome research revealed that *Lpb. plantarum* A72 upregulates the genes responsible for *sod-5* and *hsp-16.1*. This activation stimulates the production of small heat shock proteins and superoxide dismutase, curbing the generation of reactive oxygen species and facilitating the degradation of misfolded proteins. Moreover, our study elucidates for the first time how the ingestion of this strain enables *C. elegans* to suppress the synthesis of fatty acids by downregulating the *fat-6* and *lips-17* genes. This process reduces senescence and extends the lifespan of *C. elegans*.

## 4. Conclusions

The present study demonstrated that *Lpb. plantarum* A72 possesses significant antioxidant, anti-aging and longevity extension effects. These effects are due to the probiotic (including the antioxidant action) of the strain. This was demonstrated by the fact that the IC and CFS of this strain scavenged 60.14% and 87.01% of DPPH and hydroxyl radicals, respectively, while showing good growth capacity in 0–9 mM H_2_O_2_. By feeding *C. elegans* with *Lpb. plantarum* A72, we observed that *C. elegans* lifespan was extended by 25.13%, motility was enhanced 2.52-fold, and its reproductive ability was not impaired. *Lpb. plantarum* A72 reduced 34.86% of ROS and 69.52% of MDA in *C. elegans* and significantly increased the activities of related antioxidant enzymes. The strain enhanced *C. elegans* survival by 46.33% and 57.78% in high temperature and H_2_O_2_, respectively. Through transcriptomics, it was found that the strain could achieve the effect of anti-aging and prolonging lifespan of *C. elegans* by upregulating the *sod-5* and *hsp-16.1* genes and downregulating the *fat-6* and *lips-17* genes. Therefore, *Lpb. plantarum* A72 is an effective antioxidant and anti-aging strain. In the future, *Lpb. plantarum* A72 may be expected to be used in the fermentation of food and other fields. The strain provides for the development of antioxidant functional foods.

## Figures and Tables

**Figure 1 foods-13-00924-f001:**
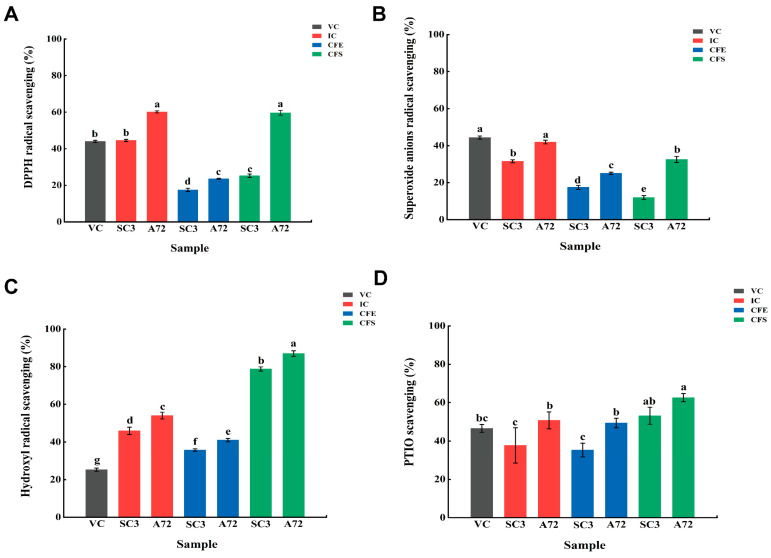
In vitro antioxidant properties of *Lpb. plantarum* SC3 and A72. (**A**) DPPH radical scavenging; (**B**) superoxide anion radical scavenging; (**C**) hydroxyl radical scavenging; (**D**) PTIO radical scavenging. (IC: intact cells; CFE: cell-free extracts; CFS: cell-free supernatant; VC: 0.1 mg/mL Vitamin C. ^a–g^ Mean values with different letters in the bar are significantly different (*p* < 0.05).

**Figure 2 foods-13-00924-f002:**
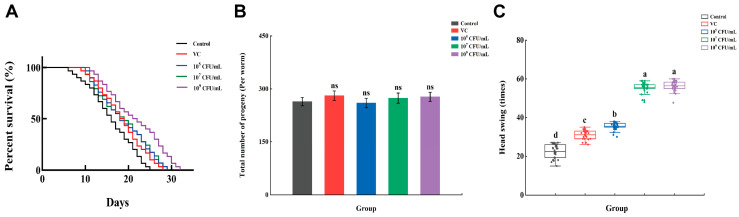
Effects of *Lpb. plantarum* A72 intact cells on the physiological conditions of *C. elegans*. (**A**) Survival curves; (**B**) total number of progeny; (**C**) head swing. (^a–d^ Mean values with different letters in the bar are significantly different (*p* < 0.05). ^ns^ no significant difference compared to the control group).

**Figure 3 foods-13-00924-f003:**
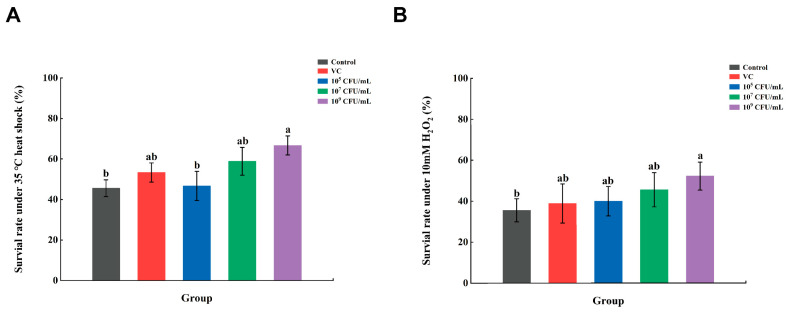
Effect of *Lpb. plantarum* A72 intact cells on *C. elegans* resistance to environmental factors. (**A**) High-temperature heat stress test; (**B**) hydrogen peroxide oxidative stress test. (^a–b^ Mean values with different letters in the bar are significantly different (*p* < 0.05)).

**Figure 4 foods-13-00924-f004:**
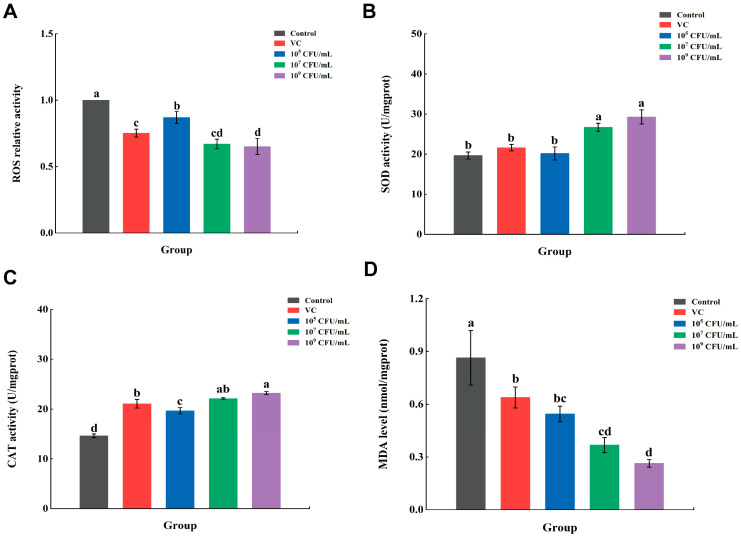
Effect of *Lpb. plantarum* A72 intact cells on the antioxidant defense system in *C. elegans*. (**A**) ROS accumulation; (**B**) SOD activity; (**C**) CAT activity; (**D**) MDA level. (^a–d^ Mean values with different letters in the bar are significantly different (*p* < 0.05)).

**Figure 5 foods-13-00924-f005:**
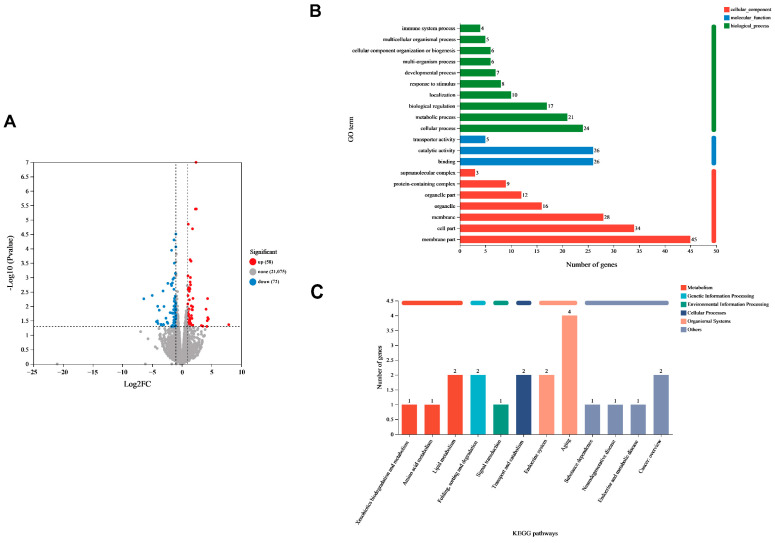
Differential gene expression of *C. elegans* induced by 10^9^ CFU/mL *Lpb. plantarum* A72. (**A**) Number of differentially expressed genes (DEGs) that were upregulated (red) and downregulated (blue) in the *Lpb. plantarum* A72-treated or control groups. Experiments were performed using three replicates per group (n = 3); (**B**) Gene ontology (GO) annotation of DEGs. The horizontal coordinates indicate the total number of differentially expressed genes annotated to GO secondary taxonomic function. The vertical coordinates indicate GO secondary taxonomic names. The same column color represents the same GO primary taxonomic classification; (**C**) Kyoto encyclopedia of genes and genomes (KEGG) annotation of DEGs. The horizontal coordinates indicate the names of KEGG metabolic pathways. The vertical coordinates indicate the total number of differentially expressed genes annotated to the pathway. The same column color represents the same KEGG metabolic pathway classification.

**Figure 6 foods-13-00924-f006:**
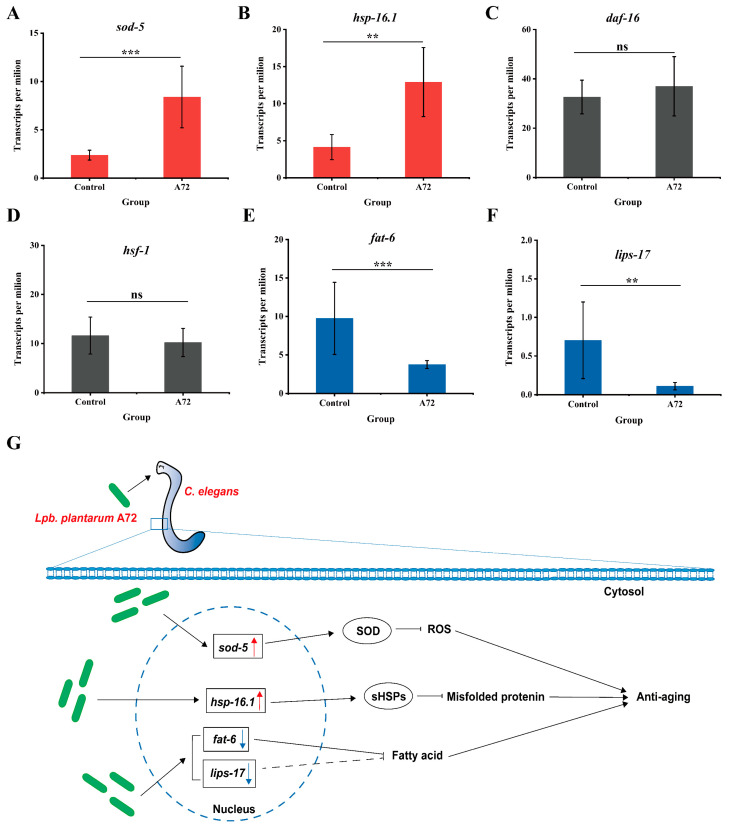
Mechanistic resolution of *C. elegans* lifespan extension by *Lpb. plantarum* A72. (**A**) Differential expression of *sod-5* genes; (**B**) differential expression of *hsp-16.1* genes; (**C**) differential expression of *daf-16* genes; (**D**) differential expression of *hsf-1* genes; (**E**) differential expression of *fat-6* genes; (**F**) differential expression of *lips-17* genes; (**G**) visualization of *Lpb. plantarum* A72 on *C. elegans* lifespan extension. Circles represent proteins. Boxes represent genes. Upward and downward arrows indicate up- and downregulation of genes, respectively. Solid lines with arrows indicate facilitation. Truncated solid lines indicate inhibition. Truncated dashed lines indicate possible inhibition. (**, *** significance of differential expression of this gene in the A72 group versus control, *p* < 0.01, *p* < 0.001, respectively; ^ns^ differential expression of this gene in the A72 group was not significant with control).

**Table 1 foods-13-00924-t001:** Effect of *Lpb. plantarum* A72 intact cells on the lifespan of *C. elegans*.

Group	Average Lifespan (Days)	Percentage Increase (%)	Maximum Lifespan (Days)
Control	25.33 ± 0.47 ^d^		26
VC	27.67 ± 0.47 ^c^	9.28 ± 3.84 ^b^	28
10^5^ CFU/mL	29.00 ± 0.82 ^bc^	14.56 ± 5.13 ^ab^	30
10^7^ CFU/mL	30.33 ± 1.25 ^ab^	19.85 ± 6.72 ^ab^	32
10^9^ CFU/mL	31.67 ± 1.25 ^a^	25.13 ± 7.08 ^a^	33

Different letters represent significant differences between groups (*p* < 0.05).

## Data Availability

The original contributions presented in the study are included in the article/Appendix A, further inquiries can be directed to the corresponding authors.

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
