# Peer review of "Lactiplantibacillus plantarum A72, a Strain with Antioxidant Properties, Obtained through ARTP Mutagenesis, Affects Caenorhabditis elegans Anti-Aging"

_foods, 2024, doi:10.3390/foods13060924_

Round 1

Reviewer 1 Report

Comments and Suggestions for Authors

The study is very interesting and soundness, but there are several mistakes that must be modified.

In the introduction, Lb plantarum are mention as a sustance, and it is incorrect, is a microorganism, which contained several molecules with infinite actions.

The mention of flora must be replaced by microbiota.

The results indicate Lb platarum have positive effect over the life of C. elegans, but the conclusion must be justified as the probiotic and post biotics effect of the microorganism, as a complex process of colonization of C. elegans digestive tract and his inmune system , not simply explained by the presence of certains molecules.

Minnor: add "et al", if it correspond, and the year of references (according author instructions)

Author Response

Special thanks to you for helping improve this paper.

Question 1: In the introduction, Lb plantarum are mention as a sustance, and it is incorrect, is a microorganism, which contained several molecules with infinite actions.

Reply 1: Thank you for your comment. We have modified the description of Lpb. plantarum on Line 35 and 36.

Question 2: The mention of flora must be replaced by microbiota.

Reply 2: Thank you for your comment. We've made changes in all the places where flora is involved.

Question 3: The results indicate Lb platarum have positive effect over the life of C. elegans, but the conclusion must be justified as the probiotic and post biotics effect of the microorganism, as a complex process of colonization of C. elegans digestive tract and his inmune system, not simply explained by the presence of certains molecules.

Reply 3: Thank you for your comment. We added “These effects are due to the probiotic and post biotics effect of the microorganism.” at the conclusions part on Line 452-453.

Question 4: Minnor: add "et al", if it correspond, and the year of references (according author instructions).

Reply 4: Thank you for your comment. We added "et al." and the year of references to each area that mentions the author.

Reviewer 2 Report

Comments and Suggestions for Authors

Comments

- In the second paragraph of the introduction, the author uses the acronym LAB referring to lactic acid bacteria; however, it is not introduced previously. Every acronym needs to be explained the first time it is used.

- In the first paragraph of page 2, the name of the alga species Laminaria japonica is not written in italics; every species name should be italicized.

- The last paragraph of the introduction presents information regarding the methodology. Organizing this section of the text.

- Line 221 (Results and Discussion): The author discusses that some articles demonstrate an increase in larval longevity; however, only one article is cited.

- Important: besides highlighting the choice of the Caenorhabditis elegans experimental model, it's crucial to consider the limitations and advantages of this experimental model when discussing the results. Make it clear to the reader the implications of using this model.

- Overall, the article is well-written. The work demonstrates consistency in its results, methods, and objectives studied.

Author Response

Special thanks to you for helping improve this paper.

Question 1: In the second paragraph of the introduction, the author uses the acronym LAB referring to lactic acid bacteria; however, it is not introduced previously. Every acronym needs to be explained the first time it is used.

Reply 1: Thank you for your comment. The “(LAB)” has been supplemented after “lactic acid bacteria” on Line 36.

Question 2: In the first paragraph of page 2, the name of the alga species Laminaria japonica is not written in italics; every species name should be italicized.

Reply 2: Thank you for your comment. The “Laminaria japonica” has been italicized on Line 54.

Question 3: The last paragraph of the introduction presents information regarding the methodology. Organizing this section of the text.

Reply 3: Thank you for your comment. The introduction presents information regarding the methodology has been corrected on Line 63-71.

Question 4: Line 221 (Results and Discussion): The author discusses that some articles demonstrate an increase in larval longevity; however, only one article is cited.

Reply 4: Thank you for your comment. We added a new article (Sharma et al., 2018) that talks about “Lcb. paracasei CD4, Lim. gastricus BTM7, and Lpb. plantarum K90 enhanced the lifespan of C. elegans by 5 days. Furthermore, the probiotic and post biotics increased the lifespan of C. elegans via adherence and further colonization in the gut of the C. elegans.” on Line 228-232.

Question 5: Important: besides highlighting the choice of the Caenorhabditis elegans experimental model, it's crucial to consider the limitations and advantages of this experimental model when discussing the results. Make it clear to the reader the implications of using this model.

Reply 5: Thank you for your comment. We added a specific description of the C. elegans model on Line 48-52.

Reviewer 3 Report

Comments and Suggestions for Authors

The manuscript describes an interesting study aimed at verifying the effect of a mutated strain of L. plantarum with high antioxidant capacity on anti-aging of C. elegans. The study is well structured, the results are presented adequately and show application potential, English is good and the methodology used is appropriate, and is well described. Some observations to improve writing and optimize the impact of the results:

- Check that the genus and species of the microorganisms are always written in italics. See lines 72, 97 and 98

-line 95: what does "NGM" mean?

- the writing is very good but could be optimized if

    a) the outstanding difference of the study in relation to previous similar studies on C. elegans is clearly expressed.

    b) express whether the mutagenesis applied to the wild strain is or not a method that can limit the application of the mutated strain in fermented foods

   c) express, beyond the scientific value of the study, for what specific application in food the results obtained could be used, that is, what projection the study has

Author Response

Special thanks to you for helping improve this paper.

Question 1: Check that the genus and species of the microorganisms are always written in italics. See lines 72, 97 and 98.

Reply 1: Thank you for your comment. The “Escherichia coli” has been italicized on Line 79-80. The “E. coli” has been italicized on Line 104 and 105.

Question 2: line 95: what does "NGM" mean?

Reply 2: Thank you for your comment. To explain what does “NGM” mean, the “Nematode Growth Medium” has been supplemented before “(NGM)” on Line 102.

Question 3: the outstanding difference of the study in relation to previous similar studies on C. elegans is clearly expressed.

Reply 3: Thank you for your comment. We added relevant descriptions of the outstanding difference of our study in relation to previous similar studies on C. elegans on Line 60-62.

Question 4: express whether the mutagenesis applied to the wild strain is or not a method that can limit the application of the mutated strain in fermented foods.

Reply 4: Thank you for your comment. We added the relevant literature on the application of ARTP technology in the food sector (Xue et al., 2019) on Line 170-173.

Question 5: express, beyond the scientific value of the study, for what specific application in food the results obtained could be used, that is, what projection the study has.

Reply 5: Thank you for your comment. We added the possible applications of Lpb. plantarum A72 involved in the food sector on Line 464-467.

Round 2

Reviewer 1 Report

Comments and Suggestions for Authors

The word "microbiota"correspond only to refer to all microorganism present in the C. elegan. When yoy mention a single strain (L. pLatarum) yor refer it as: strain (as specific mutant strain of the L. plantarum specie) or bacteria or microorganism (more general word)

the incubation of C. elegant with differents fractions of L. plantarum can modified the microbiota of C. elegans, probably by probiotic action (including the antioxidant action), or post-biotic (when use free cell supernatnat or inactivated cells), with the consecuence of C. elegance health. 

I suggest the author study the differents concept used in microbiology

Author Response

Special thanks to you for helping improve this paper.

Question 1: The word "microbiota"correspond only to refer to all microorganism present in the C. elegan. When yoy mention a single strain (L. pLatarum) yor refer it as: strain (as specific mutant strain of the L. plantarum specie) or bacteria or microorganism (more general word).

Reply 1: Thank you for your comment. Sorry for misinterpreting what you meant. When we mentioned Lpb. plantarum A72 we modified it as: strain.

Question 2: the incubation of C. elegant with differents fractions of L. plantarum can modified the microbiota of C. elegans, probably by probiotic action (including the antioxidant action), or post-biotic (when use free cell supernatnat or inactivated cells), with the consecuence of C. elegance health.

Reply 2: Thank you for your comment. We've been feeding C. elegans with live Lpb. plantarum A72, so we modified its role to “These effects are due to the probiotic (including the antioxidant action) of the strain.” on Line 449 and 450.